# Risk of Ischemic Stroke Associated with Calcium Supplements and Interaction with Oral Bisphosphonates: A Nested Case-Control Study

**DOI:** 10.3390/jcm12165294

**Published:** 2023-08-14

**Authors:** Diana Barreira-Hernández, Sara Rodríguez-Martín, Miguel Gil, Ramón Mazzucchelli, Laura Izquierdo-Esteban, Alberto García-Lledó, Ana Pérez-Gómez, Antonio Rodríguez-Miguel, Francisco J. de Abajo

**Affiliations:** 1Department of Biomedical Sciences (Pharmacology), University of Alcalá (IRYCIS), 28805 Alcalá de Henares, Spain; diana.barreira@uah.es (D.B.-H.); sara.rodriguezm@uah.es (S.R.-M.); antonio.hupa@gmail.com (A.R.-M.); 2Division of Pharmacoepidemiology and Pharmacovigilance, Spanish Agency on Medicines and Medical Devices (AEMPS), 28022 Madrid, Spain; mgilg@aemps.es; 3Rheumatology Department, University Hospital “Fundación Alcorcón”, 28922 Alcorcón, Spain; ramon.mazzucchelli@salud.madrid.org; 4Stroke Unit, Department of Neurology, University Hospital “Príncipe de Asturias”, 28805 Alcalá de Henares, Spain; lauraizes@hotmail.com; 5Department of Cardiology, University Hospital “Príncipe de Asturias”, 28805 Alcalá de Henares, Spain; alberto.garcia-lledo@uah.es; 6Department of Medicine, University of Alcalá, 28805 Alcalá de Henares, Spain; apgomez@salud.madrid.org; 7Department of Rheumatology, University Hospital “Príncipe de Asturias”, 28805 Alcalá de Henares, Spain; 8Clinical Pharmacology Unit, University Hospital “Príncipe de Asturias”, 28805 Alcalá de Henares, Spain

**Keywords:** calcium supplements, oral bisphosphonates, vitamin D, ischemic stroke, cardioembolic ischemic stroke

## Abstract

Conflicting results about the association of calcium supplements (CS) with ischemic stroke (IS) have been reported. We tested this hypothesis by differentiating between CS alone (CaM) and CS with vitamin D (CaD) and between cardioembolic and non-cardioembolic IS. We examined the potential interaction with oral bisphosphonates (oBs). A nested case-control study was carried out. We identified incident IS cases aged 40–90 and randomly sampled five controls per case matched by age, sex, and index date. Current users were compared to non-users. An adjusted odds ratios (AOR) and 95% CI were computed through conditional logistic regression. Only new users were considered. We included 13,267 cases (4400 cardioembolic, 8867 non-cardioembolic) and 61,378 controls (20,147 and 41,231, respectively). CaM use was associated with an increased risk of cardioembolic IS (AOR = 1.88; 95% CI: 1.21–2.90) in a duration-dependent manner, while it showed no association with non-cardioembolic IS (AOR = 1.05; 95% CI: 0.74–1.50); its combination with oBs increased the risk of cardioembolic IS considerably (AOR = 2.54; 95% CI: 1.28–5.04), showing no effect on non-cardioembolic. CaD use was not associated with either cardioembolic (AOR = 1.08; 95% CI: 0.88–1.31) or non-cardioembolic IS (AOR = 0.98; 95% CI: 0.84–1.13) but showed a small association with cardioembolic IS when combined with oBs (AOR = 1.35; 95% CI: 1.03–1.76). The results support the hypothesis that CS increases the risk of cardioembolic IS, primarily when used concomitantly with oBs.

## 1. Introduction

Current guidelines for the prevention and treatment of osteoporosis [1,2,3] recommend a daily calcium intake of between 700 to 1200 mg and advise a daily dose of 800 IU of vitamin D (cholecalciferol) in postmenopausal women and older men (≥50 years old) to improve bone mineral density [4]. Achieving the recommended intake through diet alone is not always possible and physicians often prescribe calcium supplementation, either in monotherapy (CaM) or in combination with vitamin D (CaD). However, the use of calcium supplements is under debate as they may have deleterious effects on the vascular system, though results are conflicting [5,6,7,8].

So far, several studies have assessed the association between the use of calcium supplements and the occurrence of cerebrovascular events [9,10,11,12,13,14]. However, few distinguished between ischemic and hemorrhagic stroke, and among those focusing on ischemic stroke (IS), none distinguished between the two main pathophysiological subtypes (cardioembolic vs. non-cardioembolic), which may be critical as the biological mechanisms underlying the potential association of calcium supplements with each subtype may substantially differ. Recently, our group reported a specific increased risk of cardioembolic IS associated with the use of oral bisphosphonates [15], suggesting an interaction between oral bisphosphonates and calcium supplements, while there was hardly any effect on non-cardioembolic IS. Bearing this in mind, we carried out the present study with a twofold aim: (1) to assess the association of calcium supplements, distinguishing between CaM and CaD, with IS overall and its two main pathophysiological subtypes, and (2) to examine the potential interaction of calcium supplements and oral bisphosphonates on IS (and its subtypes), from the perspective of calcium-supplement users.

## 2. Patients, Materials, and Methods

### 2.1. Data Source and Study Design

We performed a population-based case-control study nested in a cohort selected from BIFAP (“Base de datos para la Investigación Farmacoepidemiológica en el Ámbito Público”) over the study period from 1 January 2002 to 31 December 2015. BIFAP is a healthcare database containing pseudonymized health records of patients attended by primary care physicians (PCPs) from 9 Spanish regions (out of 17). BIFAP contains information on demographics, medical diagnostics, and drug prescriptions (indication, product name, dosage, date of prescription, and duration of treatments), among many other data [16]. This study was carried out using the 2016 version, which included 7.6 million patients with an average of 5.1 years of follow-up per subject (a total of 38.8 million person-years). Patients were enrolled in the study cohort once they fulfilled the following criteria: aged 40 to 99 years, 1-year registry with their primary care physician (PCP), and no previous record of cancer or stroke (any type). The date these criteria were met was considered the “start date”. Then, they were followed up until the occurrence of one of the following outcomes: an incident stroke (ischemic/hemorrhagic), 100 years old, a record of cancer, death, or the end of the study period, whichever came first.

### 2.2. Cases and Controls Selection

We performed initial computer research to identify all potential stroke cases using the ICPC-2 code K90 and the ICD-9-CM codes 434.x1, and 436, as well as keywords in diagnosis and comment fields. Stroke cases linked to drug abuse, vascular dissections, and aneurysms were excluded. Then, we grouped the potential cases identified into homogeneous subgroups according to the amount and type of information and extracted a random sample for each subgroup (totaling 1000 cases), and we conducted a manual revision of their complete clinical records. The case validation was carried out independently by two of the investigators (SRM and DBH), who were blinded to drug exposure, and discrepancies were solved by the whole research group (including a neurologist serving in a stroke unit (L.I.-E.) and a cardiologist (A.G.-L.). Further, we performed a second validation to identify the most probable pathophysiological subtype of IS (cardioembolic or non-cardioembolic), described elsewhere [17]. Briefly, an IS was considered cardioembolic if it met the following main criteria: (1) a note that it was cardioembolic accompanied the diagnosis of IS; or (2) a record of atrial fibrillation before or within 3 months after the index date; or (3) prescriptions of anticoagulants recorded before or within 3 months after the index date (74.1% of cardioembolic IS cases met at least two of them). A record of valvular prosthesis or mitral stenosis before the index date and prescriptions of class IC or class III antiarrhythmics recorded before or within three months after the index date, in addition to any of the criteria mentioned above, were considered as supportive information of the diagnosis of cardioembolic stroke. All IS cases not fulfilling these criteria were considered non-cardioembolic.

We considered the “index date” as the date of the first record of IS. Five controls per case individually matched with cases by exact age, sex, and index date were randomly selected from the underlying cohort following risk-set sampling. As controls arose from the underlying cohort, they fulfilled the same criteria as cases: aged 40 to 99 years, 1-year registry with their primary care physician (PCP), and no previous record of cancer or stroke. Such sampling of controls is incidence-density-based (the probability of control selection is proportional to the total person-time at risk) and allows for obtaining unbiased estimates of the underlying cohort’s rate ratios (RRs) through the ORs computed in the case-control analysis [18].

### 2.3. New-User Design

We only selected initiators of calcium supplements among cases and controls. To this end, we excluded subjects with prescriptions of calcium supplements prior to the cohort start date.

### 2.4. Exposure Definition

New users of calcium supplements were categorized as “current users” when they had at least a recorded prescription that ended within 365 days before the index date and “past users” when the prescription ended beyond 365 days before the index date. Those patients without a recorded prescription of calcium supplements before the index date were considered “non-users”. Among current users, we distinguished those who used them as monotherapy (CaM) or with vitamin D (CaD) (either in fixed-dose combinations or as separate medicinal products) and obtained the daily dose (“low” when it was less than 1000 mg of elemental calcium and “high” when it was equal to or over 1000 mg), and the treatment duration (sum of all consecutive prescriptions with a maximum accepted gap of 60 days between the end of the supply of one prescription and the start of the next one). The duration was categorized as equal to or less than 365 days and over 365 days.

### 2.5. Potential Confounding Factors

To compute the adjusted odds ratios (AORs), the following co-morbidities and other risk factors were considered as possible confounding factors using expert criteria (all of them recorded at the index date or before): Number of visits to primary care physician (PCP) during the year preceding the index date, body mass index (BMI), alcohol abuse, smoking, diabetes (recorded as such and/or by using glucose-lowering drugs as an indicator), hypertension, dyslipidemia (recorded as such and/or by using lipid-lowering drugs as an indicator), hyperuricemia (either asymptomatic or in the context of gout), peripheral artery disease (PAD), acute myocardial infarction (AMI), angina pectoris (recorded as such and/or use of nitrates), transient ischemic attack (TIA), thromboembolic disease, heart failure, chronic kidney failure, rheumatoid arthritis, and chronic obstructive pulmonary disease (COPD). All these comorbidities were recorded as such by the PCPs and further validated by the BIFAP team through the manual review of a small sample. For some comorbidities with specific drugs (diabetes, dyslipidemia, angina), we also used such drugs as disease indicators (as specified). Furthermore, current use of the following drugs was considered as a covariate: antiplatelet drugs, alpha-blockers, beta-blockers, calcium channel blockers, angiotensin-converting enzyme inhibitors, angiotensin II receptor blockers, diuretics, nonsteroidal anti-inflammatory drugs (both non-selective and COX-2 selective), paracetamol (acetaminophen), metamizole, opioids, systemic corticosteroids, proton pump inhibitors, H2-receptors antagonists, active forms of vitamin D (calcifediol, calcitriol, and alfacalcidol), hormonal replacement therapy, selective estrogen receptor modulators, strontium ranelate, oral bisphosphonates, and other agents for osteoporosis treatment (calcitonin, denosumab, and teriparatide). Atrial fibrillation and oral anticoagulant use were not considered potential confounding factors because they were part of the criteria used to define the cardioembolic IS (see Section 2.2).

### 2.6. Statistical Analysis

To estimate the association between calcium supplements and incident IS, we built a conditional logistic regression model and computed the odds ratio (OR) and its 95% confidence intervals (CIs), considering only the matching variables (exact age, sex, and index date). Further, we computed the adjusted odds ratios (AORs) by adding the confounding factors detailed above.

The potential interaction of calcium supplements with age (less than 70 and ≥70 years old), sex, background vascular risk (see below), CHA2DS2-VASc score (equal to 3 or less and greater), and current use of oral bisphosphonates was explored through stratified analyses. Background vascular risk was categorized into: (1) establish vascular disease, including patients with a record of PAD, AMI, or TIA; (2) vascular risk factors, including patients with a record of hypertension, diabetes mellitus, chronic renal failure, atrial fibrillation, dyslipidemia, current smoking, or BMI higher than 30 kg/m^2^ without any of the vascular diseases aforementioned; and (3) no vascular risk factors, the remainder.

AORs obtained in different strata were compared using the interaction test described by Altman and Bland [19] (multiplicative interaction). The potential interaction of calcium supplements with oral bisphosphonates was also examined in the additive scale [20]. To that end, we built a variable with seven categories: (1) non-use of calcium supplements and non-use of oral bisphosphonates; (2) current use of CaM alone; (3) current use of CaD alone; (4) current use of oral bisphosphonates alone; (5) current use of CaM and oral bisphosphonates; (6) current use of CaD and oral bisphosphonates; and (7) the remainder (those patients who are: (a) past users of calcium supplements and past users of oral bisphosphonates; or (b) past users of calcium supplements and non-users of oral bisphosphonates; or (c) past users of oral bisphosphonates and non-users of calcium supplements).

We used unconditional logistic regression models for the stratified analysis by variables different from the matching ones, as conditional models produced unstable estimates.

We had missing values for BMI (30%) and smoking (46%) that were identified in specific categories within the variables. In a sensitivity analysis, however, we assessed their impact on the main results by applying multiple imputations by chained equation models (MICE) [21].

We only computed the AOR when there were more than five exposed cases per category. A *p*-value of less than 0.05 was considered statistically significant. No adjustment of multiple testing was carried out. All analyses were performed with STATA/SE 15 (StataCorp, College Station, TX, USA).

### 2.7. Ethical Aspects

The study was conducted according to the principles of the Helsinki Declaration (2013), as well as Spanish and European laws. The BIFAP Scientific Committee approved the study protocol of a bigger project on atherothrombotic events and the use of different drugs, which included the current one, on 26 May 2016 (project #04/2016). The present study was approved by the Research Ethics Committee of the University Hospital Príncipe de Asturias (Ref #CAL-BIS-CACO, #EOm 05/2022) on 29 March 2022. The data used were fully pseudonymized, and the Committee granted a waiver for the informed consent.

## 3. Results

A total of 13,267 incident IS cases (4400 classified as cardioembolic and 8867 as non-cardioembolic), and 61,378 matched controls (20,147 with cardioembolic cases and 41,231 with non-cardioembolic cases) were included (Figure 1).

Baseline characteristics of cardioembolic and non-cardioembolic IS cases and their respective controls are shown in Table 1 (see Appendix A for IS cases overall and their controls). As expected, cases had a higher prevalence of comorbidities and use of drugs for cardiovascular conditions than controls. Among controls, we explored the prevalence of comorbidities, cardiovascular risk factors, and co-medication of current users of CaM and CaD as compared to non-users. As shown in Appendix A and Appendix A (Appendix A), CaM and CaD users presented a higher prevalence of comorbidities and co-medications than non-users, with hardly any difference between CaM and CaD users.

Cardioembolic IS cases were older (mean (SD) age: 76.5 ± 11.39 years old) than non-cardioembolic IS cases (mean (SD) age: 73.2 ± 12.85) and also had a higher prevalence of comorbidities and co-medications.

### 3.1. CaM and Risk of Ischemic Stroke Overall and by Pathophysiological Subtype

Overall, IS cases presented a higher proportion of current users of CaM than controls (0.58% and 0.41%, respectively), yielding an AOR of 1.25 (95% CI: 0.95–1.63). The increased AOR was only significant with durations longer than 1 year (AOR_>1 year_ = 1.78; 95% CI: 1.10–2.86 vs. AOR_≤1 year_ = 1.07; 95% CI: 077–149). No relationship with calcium daily dose was found (Table 2).

We found a dramatic difference when the association of CaM with IS was examined by its main pathophysiological subtypes. For cardioembolic IS, the AOR was 1.88 (95% CI: 1.21–2.90), with a significant trend with duration (AOR_≤1 year_ = 1.60; 95% CI: 0.94–2.70, and AOR_>1 year_ = 2.73; 95% CI: 1.25–5.94; *p* for trend = 0.019) and no significant change with calcium daily dose (AOR_Low dose (<1000 mg/d)_ = 2.28; 95% CI: 1.29–4.00, and AOR_High dose (≥1000 mg/d)_ = 1.63; 95% CI: 0.70–3.81) (Table 3). By contrast, for non-cardioembolic IS, the AOR was 1.05 (95% CI: 0.74–1.50), and no change was observed with duration or daily dose (Table 4).

In the stratified analyses by subgroups of age, sex, background vascular risk, and CHA2DS2-VASc score, we observed that the current use of CaM was associated with an increased risk of cardioembolic IS in almost all the strata, with no evidence of statistical interaction with any of those variables. No increased risk associated with non-cardioembolic IS was observed among current users of CaM in any subgroup (Figure 2).

### 3.2. CaD and Risk of Ischemic Stroke Overall and by Pathophysiological Subtype

Current use of CaD showed a higher prevalence among IS cases (4.09%) than among controls (3.60%), leading to an AOR of 1.02 (95% CI: 0.91–1.14) compared to non-users. According to treatment duration (1 year or less, and over 1 year), current use of CaD showed an AOR of 0.96 (95% CI: 0.84–1.09) and 1.19 (95% CI: 0.98–1.43), respectively. By daily dose of calcium, the AORs were 0.87 (95% CI: 0.73–1.02) and 1.19 (95% CI: 1.01–1.40), for less than 1000 mg/d and 1000 mg/d or more, respectively (Table 2).

We did not find a significant increased risk for any IS subtype. For cardioembolic IS, the AOR was 1.08 (95% CI: 0.88–1.31), and no significant change was observed with duration or daily dose (Table 3). For non-cardioembolic IS, the AOR was 0.98 (95% CI: 0.84–1.13), and no change was observed with the daily dose of calcium; however, with a duration longer than 365 days, an AOR of 1.40 (95% CI: 1.11–1.77) was found (Table 4). There was no evidence of the interaction of CaD with age, sex, background vascular risk, and CHA2DS2-VASc score for both cardioembolic and non-cardioembolic IS (Figure 3).

### 3.3. Interaction of Calcium Supplements with Oral Bisphosphonates

The current use of CaM was associated with an increased risk of cardioembolic IS in the stratum of patients who concomitantly used oral bisphosphonates (AOR = 3.20; 95% CI: 1.21–8.47), while there was no significant association among non-users of oral bisphosphonates (AOR = 1.40; 95% CI: 0.78–2.53) (*p* for interaction = 0.1541). For non-cardioembolic IS, the current use of CaM was not associated with an increased risk regardless of the concomitant use of oral bisphosphonates. The use of CaD did not show a statistical interaction with oral bisphosphonates in any subtype of IS (Figure 4).

The potential interaction between CaM or CaD and oral bisphosphonates on each IS subtype was also examined in the additive scale. For cardioembolic IS, we found an increased risk associated with the concomitant use of CaM and oral bisphosphonates as compared to the non-use of any (AOR = 2.54; 95% CI: 1.28–5.04), while no significant increased risk was observed when either CaM or oral bisphosphonates were used singly (AOR = 1.47; 95% CI: 0.81–2.68; and AOR = 1.08; 95% CI: 0.76–1.56, respectively). Also, the concomitant use of CaD with oral bisphosphonates was associated with a significantly increased risk as compared to the non-use of any (AOR = 1.35; 95% CI: 1.03–1.76), but this was much smaller than with CaM. On the contrary, the combined use of CaM or CaD with oral bisphosphonates did not increase the risk of non-cardioembolic IS (Figure 5).

## 4. Discussion

The results of the present study suggest that the use of calcium supplements without vitamin D (CaM) increases the risk of cardioembolic IS without affecting the risk of non-cardioembolic IS. Such effect was observed with both low and high daily doses of calcium supplements and with any duration of use, though the highest risk appeared with a duration longer than 1 year. No interaction with sex, age, background vascular risk, or CHA2DS2-VASc score was detected. Notably, the results suggest the existence of an interaction with oral bisphosphonates, so the increased risk of cardioembolic IS associated with CaM was only observed when they were used combined with oral bisphosphonates, but not when both drugs were used singly.

The use of calcium supplements with vitamin D (CaD) was neither associated with IS overall nor with any of the main pathophysiological subtypes. However, a moderately increased risk of non-cardioembolic IS was observed with durations longer than 1 year. No interaction with sex, age, background vascular risk, or CHA2DS2-VASc score was found. A moderately increased risk of cardioembolic IS was detected when oral bisphosphonates and CaD were used concomitantly, but it was much smaller than the one found with CaM.

Several meta-analyses of randomized clinical trials (RCTs) that examined the association between calcium supplementation and the risk of cardiovascular events and stroke have been published, showing conflicting results. In 2010 and 2011, Bolland et al. [9,10] reported an association between the use of calcium supplements with an increased risk of myocardial infarction and stroke. In 2013, Mao P.-J. et al. [22] reported that there might be an increased risk of major cardiovascular events (myocardial infarction and coronary heart disease) associated with the use of calcium supplements with or without vitamin D. However, they did not find a statistically significant increased risk of stroke. In 2015, Lewis et al. [23] and in 2016, Chung et al. [11] reported that the use of calcium supplements was not associated with cardiovascular risk in generally healthy adults. In 2020, Yang et al. [24], in a meta-analysis including 26 prospective cohort studies and 16 RCTs, concluded that calcium intake from diet did not increase the risk of cardiovascular events or stroke, whereas calcium supplements could increase the risk of coronary heart disease, especially myocardial infarction, but not the risk of stroke. In 2021, Myung et al. [25] concluded that calcium supplements were associated with a significantly increased risk of cardiovascular events by about 15% and a non-significant increased risk of stroke by about 13% in healthy postmenopausal women. Finally, in 2023, Huo et al. [26] found with CaM a non-significant trend for both acute myocardial infarction (RR = 1.15; 95% CI: 0.88–1.51) and stroke (1.15; 95% CI: 0.90–1.46), while for CaD the trend was much smaller for AMI (1.09; 95% CI: 0.95–1.25) and almost null for stroke (RR = 1.02; 95% CI: 0.89–1.17).

All in all, the evidence from RCTs of the increased risk of using calcium supplements is weak either used alone or combined with vitamin D. Nevertheless, it is important to note that the duration was not analyzed. Also, as far as it concerns stroke, no consideration was given to the different pathophysiological types of strokes (hemorrhagic vs. ischemic, and among ischemic stroke, cardioembolic vs. non-cardioembolic). Also, no information is provided about the concomitant use of bisphosphonates. Our data suggest that all these points matter: duration of treatment, a precise definition of the type of ischemic stroke, and the concomitant use of bisphosphonates. Thus, in our view, the possible association of calcium supplements with ischemic stroke, particularly when given without vitamin D, is open.

Cardioembolic IS occurs as a consequence of endothelial injury, stasis, and hypercoagulability at the cardiac level [27]. Atrial fibrillation is believed to be the most prevalent cause of cardioembolic IS, although growing evidence suggests that atrial thrombogenesis can occur without a rhythm disorder. Kamel et al. [28] have proposed a new model in which an underlying atrial cardiopathy (with or without atrial fibrillation) would be the main biological substrate for thrombogenesis and posterior embolization. The mechanism by which CaM may increase the risk of cardioembolic IS is uncertain.

Calcium is a well-known coagulation factor, and it has been postulated that serum peak levels reached after the intake of calcium supplements may induce a transient hypercoagulation state that would promote thrombotic events [29]. However, this acute effect could not account for the null effect found with calcium supplements associated with vitamin D, nor would it explain the greater effect found with longer treatments and the fact that non-cardioembolic IS remained materially unaffected. The suggested interaction of calcium supplements with oral bisphosphonates is an interesting finding that may provide some clues. Our group recently reported a specific increased risk of cardioembolic IS with oral bisphosphonates [15], which was clearly duration-dependent. Based on the model proposed by Kamel et al. [28] for cardioembolism of atrial origin, we postulated that the long-term use of bisphosphonates might induce an atrial cardiopathy, associated or not with atrial fibrillation, which ultimately could be the primary biological substrate for thrombogenesis. Following this reasoning, the interaction with calcium supplements, especially when used without vitamin D, suggests that calcium and bisphosphonates may potentiate each other to promote long-run atrial cardiopathy, which could be the leading cause of the observed increased risk of cardioembolic IS. In this context, vitamin D may be acting as a preventive factor, which is consistent with abundant evidence suggesting a cardioprotective effect of vitamin D [30,31,32,33,34,35,36,37], including the recently published D-Health trial [38]. Nevertheless, two other major trials (VITAL [39] and VIDA [40]) did not show a preventive cardiovascular effect in a primary prevention setting. Interestingly, in the Finnish Vitamin D trial, Virtanen et al. [41] found that vitamin D_3_ supplementation was associated with a reduced risk of atrial fibrillation.

The main strengths of our study are: (1) the PCPs collected the clinical information prospectively and filled the prescriptions using the computer system, making the misclassification of the exposure highly unlikely; (2) controls were randomly sampled from the source population, which prevents a control-selection bias; and (3) only new users were considered, avoiding a prevalent-user bias. Among the limitations are: (1) the present study is observational, and residual confounding due to unmeasured or unknown confounders is still possible; (2) treatment adherence, as in any clinical study, is not guaranteed; (3) despite the validation effort to determine the most probable pathophysiological subtype of IS (cardioembolic and non-cardioembolic), there may still be some misclassification.

## 5. Conclusions

The results of the present study suggest that the long-term use of calcium supplements without vitamin D increases the risk of cardioembolic IS, which seems particularly great when used in combination with oral bisphosphonates. Our results also suggest a moderately increased risk of cardioembolic IS when calcium supplements with vitamin D are used combined with oral bisphosphonates, but of much lesser magnitude. A moderately increased risk of non-cardioembolic ischemic stroke associated with the long-term use of calcium supplements with vitamin D is also suggested. These data urge us to reconsider the use of calcium supplements in preventing and treating osteoporosis [42,43], in particular as a complement to antiresorptive therapy with oral bisphosphonates.

## Figures and Tables

**Figure 1 jcm-12-05294-f001:**
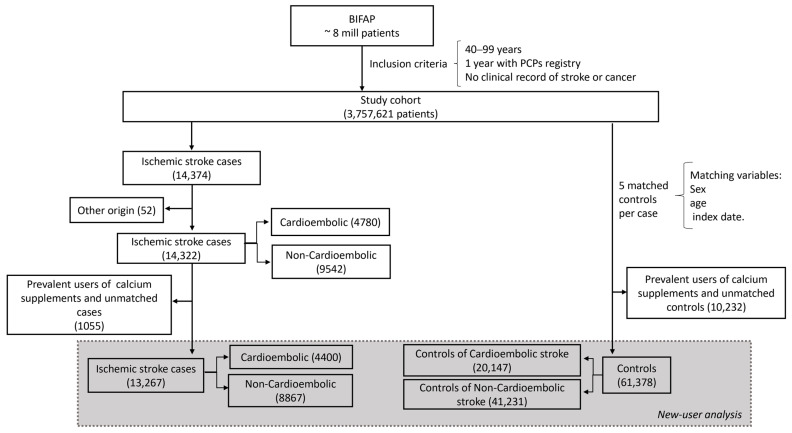
Patient selection flowchart.

**Figure 2 jcm-12-05294-f002:**
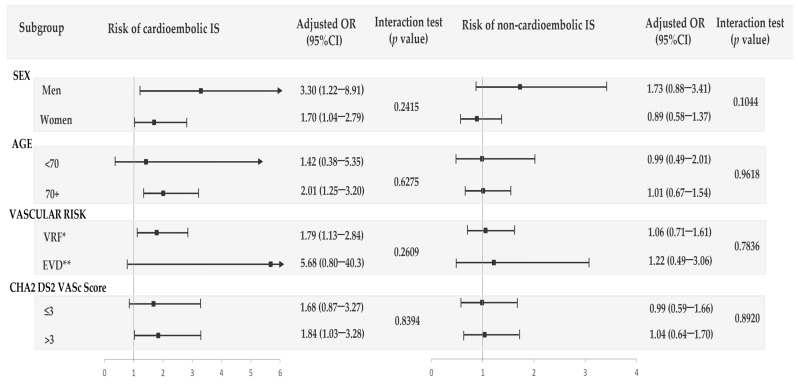
Use of calcium supplements without vitamin D (CaM) and risk of the two main subtypes of IS (non-cardioembolic and cardioembolic) by sex, age, vascular risk, and CHA2 DS2-VASc score. * VRF (vascular risk factors) included patients with at least one vascular risk factor: hypertension, renal failure, dyslipidemia, diabetes mellitus, atrial fibrillation, current smoking or body mass index higher than 30 kg/m^2^. ** EVD (establish vascular disease) included those with at least one record of peripheral artery disease, acute myocardial infarction, or transient ischemic attack.

**Figure 3 jcm-12-05294-f003:**
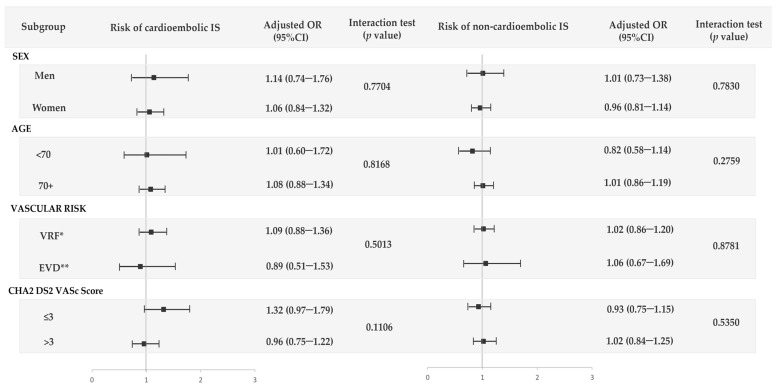
Use of calcium supplements with vitamin D (CaD) and risk of the two main subtypes of IS (non-cardioembolic and cardioembolic) by sex, age, vascular risk, and CHA2DS2-VASc score. * VRF (vascular risk factors) included patients with at least one vascular risk factor: hypertension, renal failure, dyslipidemia, diabetes mellitus, atrial fibrillation, current smoking or body mass index higher than 30 kg/m^2^. ** EVD (establish vascular disease) included those with at least one record of peripheral artery disease, acute myocardial infarction, or transient ischemic attack.

**Figure 4 jcm-12-05294-f004:**
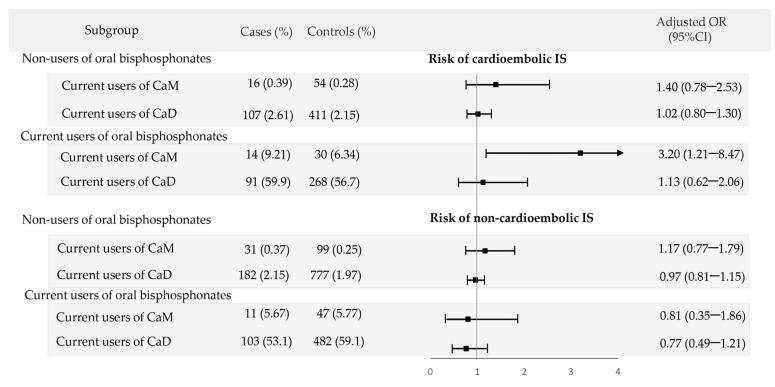
Risk of cardioembolic and non-cardioembolic ischemic stroke associated with the use of calcium supplements alone (CaM) or with vitamin D (CaD) stratified by the concomitant use of oral bisphosphonates.

**Figure 5 jcm-12-05294-f005:**
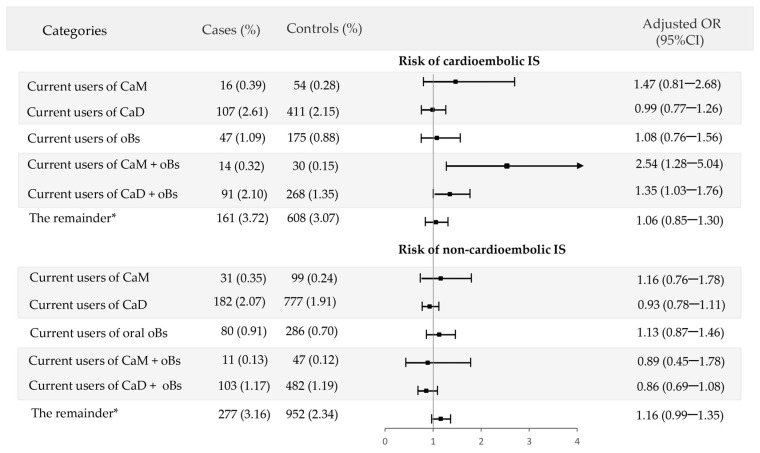
Interaction between calcium supplements with vitamin D (CaD) or without (CaM) and oral bisphosphonates (oBs) in the additive scale for the two pathophysiological subtypes of ischemic stroke (cardioembolic and non-cardioembolic). * The remainder are: (a) Past users of calcium supplements and past users of oral bisphosphonates; (b) past users of calcium supplements and non-users of oral bisphosphonates; or (c) past users of oral bisphosphonates and non-users of calcium supplements.

**Table 1 jcm-12-05294-t001:** Characteristics of cardioembolic and non-cardioembolic ischemic stroke (IS) cases and their respective controls.

	Cardioembolic IS		Non-Cardioembolic IS	
	CASES N = 4400	CONTROLS N = 20,147	CASES N = 8867	CONTROLS N = 41,231
Age, mean (SD), years	76.5 (±11.39)	76.3 (±11.56)	73.2 (±12.85)	72.8 (±12.97)
Men, (%)	2110 (47.95)	10,346 (51.35)	4911 (55.39)	24,145 (58.56)
Number of visits to PCP, (%)				
<6	585 (13.30)	5290 (26.26)	1831 (20.65)	12,771 (30.97)
6–15	1459 (33.16)	7478 (37.12)	3463 (39.05)	15,595 (37.82)
16–24	1073 (24.39)	3942 (19.57)	1987 (22.41)	7100 (17.22)
>24	1283 (29.16)	3437 (17.06)	1586 (17.89)	5765 (13.98)
BMI, No. (%), kg·m^−2^				
<25	642 (14.59)	2776 (13.78)	1185 (13.36)	5314 (12.89)
25–29.9	1348 (30.64)	5947 (29.52)	2567 (28.95)	11,913 (28.89)
30–34.9	842 (19.14)	3412 (16.94)	1529 (17.24)	6726 (16.31)
35–39.9	218 (4.95)	823 (4.08)	465 (5.24)	1774 (4.30)
>40	88 (2.00)	222 (1.10)	140 (1.58)	455 (1.10)
Unknown	1262 (28.68)	6967 (34.58)	2981 (33.62)	15,049 (36.50)
Smoking, (%)				
Non-smoker	1630 (37.05)	6845 (33.98)	2661 (30.01)	12,582 (30.52)
Current smoker	532 (12.09)	2140 (10.62)	1659 (18.71)	5297 (12.85)
Past smoker	356 (8.09)	1129 (5.60)	574 (6.47)	2365 (5.74)
Unknown	1882 (42.77)	10,033 (49.80)	3973 (44.81)	20,987 (50.90)
Alcohol abuse ^a^	112 (2.55)	308 (1.53)	299 (3.37)	710 (1.72)
Diabetes ^b^	1154 (26.23)	3979 (19.75)	2715 (30.62)	7728 (18.74)
Hyperuricemia				
Asymptomatic	429 (9.75)	1500 (7.45)	633 (7.14)	2953 (7.16)
Gout	256 (5.82)	834 (4.14)	419 (4.73)	1727 (4.19)
Hypertension	2889 (65.66)	11,242 (55.80)	5355 (60.39)	20,676 (50.15)
Dyslipidemia ^c^	2031 (46.16)	8050 (39.96)	3810 (42.97)	15,726 (38.14)
Peripheral artery disease	241 (5.48)	539 (2.68)	431 (4.86)	1011 (2.45)
Ischemic heart disease				
Acute myocardial infarction	375 (8.52)	746 (3.70)	445 (5.02)	1526 (3.70)
Angina pectoris ^d^	515 (11.70)	1332 (6.61)	667 (7.52)	2525 (6.12)
TIA	295 (6.70)	478 (2.37)	442 (4.98)	852 (2.07)
Atrial fibrillation	1903 (43.25)	1682 (8.35)	48 (0.54)	2912 (7.06)
Thromboembolic disease	132 (3.00)	332 (1.65)	152 (1.71)	635 (1.54)
Heart failure	612 (13.91)	990 (4.91)	366 (4.13)	1663 (4.03)
Chronic renal failure	289 (6.57)	817 (4.06)	442 (4.98)	1431 (3.47)
Rheumatoid arthritis	27 (0.61)	152 (0.75)	52 (0.59)	263 (0.64)
COPD	407 (9.25)	1606 (7.97)	744 (8.39)	3053 (7.40)
Background vascular risk				
No risk factors/diseases	376 (8.55)	4182 (20.76)	1399 (15.78)	9969 (24.18)
Risk factors only	3194 (72.59)	14,325 (71.10)	6259 (70.59)	28,068 (68.07)
Established vascular disease	830 (18.86)	1640 (8.14)	1209 (13.63)	3194 (7.75)
CHA2DS2-VASc score				
Mean (±SD)	3.39 (±1.63)	2.90 (±1.50)	2.89 (±1.62)	2.51 (±1.59)
≤3	2212 (50.27)	12,727 (63.17)	5539 (62.47)	29,239 (70.92)
>3	2188 (49.73)	7420 (36.83)	3328 (37.53)	11,992 (29.08)
**Current use of:**				
Antiplatelet drugs	1178 (26.77)	3484 (17.29)	2346 (26.46)	6487 (15.73)
NSAIDs	334 (7.59)	2000 (9.93)	856 (9.65)	3818 (9.26)
Oral anticoagulants drugs	1013 (23.02)	1255 (6.23)	29 (0.33)	2111 (5.12)
Paracetamol (Acetaminophen)	807 (18.34)	3533 (17.54)	1354 (15.27)	6128 (14.86)
Metamizole	223 (5.07)	824 (4.09)	404 (4.56)	1423 (3.45)
Opioids	231 (5.25)	891 (4.42)	394 (4.44)	1498 (3.63)
Proton pump inhibitors	1608 (36.55)	5698 (28.28)	2625 (29.60)	10,225 (24.80)
H2-receptor antagonists	92 (2.09)	368 (1.83)	218 (2.46)	677 (1.64)
Corticosteroids	107 (2.43)	310 (1.54)	161 (1.82)	633 (1.54)
ACEIs	1028 (23.36)	3618 (17.96)	1777 (20.04)	7007 (16.99)
ARBs	846 (19.23)	3291 (16.33)	1485 (16.75)	5907 (14.33)
Calcium channel blockers	735 (16.70)	2655 (13.18)	1317 (14.85)	4806 (11.66)
β-Blockers	1066 (24.23)	1935 (9.60)	967 (10.91)	3798 (9.21)
α-Blockers	118 (2.68)	494 (2.45)	227 (2.56)	896 (2.17)
Diuretics	1195 (27.16)	2913 (14.46)	1172 (13.22)	5082 (12.33)
Active forms of vitamin D	20 (0.45)	109 (0.54)	53 (0.60)	170 (0.41)
Bisphosphonates	136 (3.09)	477 (2.37)	163 (1.84)	847 (2.05)
Hormonal replacement therapy ^e^	2 (0.05)	13 (0.06)	9 (0.10)	52 (0.13)
SERM	7 (0.16)	31 (0.15)	10 (0.11)	69 (0.17)
Strontium ranelate	10 (0.23)	35 (0.17)	14 (0.16)	43 (0.10)
Calcitonin	3 (0.07)	15 (0.07)	5 (0.06)	47 (0.11)
Denosumab	3 (0.07)	10 (0.05)	3 (0.03)	14 (0.03)
Teriparatide	4 (0.09)	13 (0.06)	5 (0.06)	20 (0.05)

Abbreviations: ACEIs: angiotensin-converting enzyme inhibitors; ARBs: angiotensin II receptor blockers; BMI: body mass index; COPD: chronic obstructive pulmonary disease; IS: ischemic stroke; NSAIDs: nonsteroidal anti-inflammatory drugs; PCP: primary care physician; SERM: selective estrogen receptor modulators; TIA: transient ischemic attack. ^a^ When the general practitioner recorded an excessive consumption of alcohol. ^b^ Recorded as such or when patients were using glucose-lowering drugs. ^c^ Recorded as such or when patients were using lipid-lowering drugs. ^d^ Recorded as such and/or use of nitrates. ^e^ Including tibolone.

**Table 2 jcm-12-05294-t002:** Risk of ischemic stroke associated with the use of calcium supplements.

Overall IS	Cases (%) N = 13,267	Controls (%) N = 61,378	Unadjusted OR * (95% CI)	Adjusted OR ^†^ (95% CI)
Non-users	12,225 (92.15)	57,434 (93.57)	1 (Ref.)	1 (Ref.)
Recency of use				
Current users of CaM	77 (0.58)	252 (0.41)	1.36 (1.05–1.76)	1.25 (0.95–1.63)
Current users of CaD	543 (4.09)	2211 (3.60)	1.09 (0.99–1.21)	1.02 (0.91–1.14)
Past users of CaM/CaD	422 (3.18)	1481 (2.41)	1.28 (1.14–1.43)	1.14 (1.00–1.29)
Duration				
Among current users of CaM				
≤1 year	50 (0.38)	186 (0.30)	1.19 (0.87–1.63)	1.08 (0.78–1.51)
>1 year	27 (0.20)	66 (0.11)	1.84 (1.17–2.88)	1.74 (1.08–2.79)
Among current users of CaD				
≤1 year	368 (2.70)	1571 (2.56)	1.02 (0.91–1.15)	0.96 (0.84–1.09)
>1 year	185 (1.39)	640 (1.04)	1.28 (1.08–1.51)	1.19 (0.98–1.43)
Daily dose of calcium				
Among current users of CaM				
Low dose (<1000 mg/d)	41 (0.31)	145 (0.24)	1.24 (0.87–1.76)	1.22 (0.85–1.76)
High dose (≥1000 mg/d)	23 (0.17)	58 (0.09)	1.81 (1.12–2.95)	1.38 (0.82–2.30)
Unknown	13 (0.10)	49 (0.08)	1.19 (0.64–2.19)	1.13 (0.60–2.12)
Among current users of CaD				
Low dose (<1000 mg/d)	216 (1.63)	1022 (1.67)	0.93 (0.80–1.08)	0.87 (0.73–1.02)
High dose (≥1000 mg/d)	246 (1.85)	862 (1.40)	1.28 (1.11–1.48)	1.19 (1.01–1.40)
Unknown	81 (0.61)	327 (0.53)	1.12 (0.87–1.43)	1.07 (0.83–1.39)

* Adjusted only for matching factors (age, sex, and calendar year). ^†^ Adjusted for matching factors (age, sex, and calendar year) and the potential confounding factors shown in Table 1. Except for atrial fibrillation and oral anticoagulant drugs.

**Table 3 jcm-12-05294-t003:** Risk of cardioembolic ischemic stroke associated with the use of calcium supplements.

Cardioembolic Stroke	Cases (%) N = 4400	Controls (%) N = 20,147	Unadjusted OR * (95% CI)	Adjusted OR ^†^ (95% CI)
Non-users	3995 (90.80)	18,687 (92.75)	1 (Ref.)	1 (Ref.)
Recency of use				
Current users of CaM	34 (0.77)	90 (0.45)	1.67 (1.12–2.49)	1.88 (1.21–2.90)
Current users of CaD	224 (5.09)	788 (3.91)	1.27 (1.09–1.49)	1.08 (0.88–1.31)
Past users of CaM/CaD	147 (3.34)	582 (2.89)	1.14 (0.94–1.37)	0.93 (0.74–1.16)
Duration				
Among current users of CaM				
≤1 year	21 (0.48)	70 (0.35)	1.32 (0.80–2.15)	1.60 (0.94–2.70)
>1 year	13 (0.30)	20 (0.10)	2.95 (1.46–5.96)	2.73 (1.25–5.94)
Among current users of CaD				
≤1 year	159 (3.61)	531 (2.64)	135 (1.12–1.62)	1.17 (0.94–1.45)
>1 year	65 (1.48)	257 (1.28)	1.12 (0.85–1.48)	0.88 (0.63–1.21)
Daily dose of calcium				
Among current users of CaM				
Low dose (<1000 mg/d)	21 (0.48)	50 (0.25)	1.80 (1.08–3.01)	2.28 (1.29–4.00)
High dose (≥1000 mg/d)	9 (0.20)	20 (0.10)	2.09 (0.95–4.61)	1.63 (0.70–3.81)
Unknown	4 (0.09)	20 (0.10)	0.89 (0.30–2.63)	1.18 (0.38–3.62)
Among current users of CaD				
Low dose (<1000 mg/d)	90 (2.05)	375 (1.86)	1.06 (0.83–1.34)	0.90 (0.69–1.18)
High dose (≥1000 mg/d)	101 (2.30)	299 (1.48)	1.53 (1.22–1.93)	1.30 (0.99–1.71)
Unknown	33 (0.75)	114 (0.57)	1.33 (0.90–1.96)	1.08 (0.70–1.69)

* Adjusted only for matching factors (age, sex, and calendar year). ^†^ Adjusted for matching factors (age, sex, and calendar year) and the potential confounding factors shown in Table 1. Except for atrial fibrillation and oral anticoagulant drugs.

**Table 4 jcm-12-05294-t004:** Non-cardioembolic ischemic stroke associated with use of calcium supplements.

Non-Cardioembolic Stroke	Cases (%) N = 8867	Controls (%) N = 41,231	Unadjusted OR * (95% CI)	Adjusted OR ^†^ (95% CI)
Non-users	8230 (92.82)	38,747 (93.98)	1 (Ref.)	1 (Ref.)
Recency of use				
Current users of CaM	43 (0.48)	162 (0.39)	1.19 (0.85–1.67)	1.05 (0.74–1.50)
Current users of CaD	319 (3.60)	1423 (3.45)	0.99 (0.88–1.13)	0.98 (0.84–1.13)
Past users of CaM/CaD	275 (3.10)	899 (2.18)	1.37 (1.19–1.58)	1.26 (1.07–1.48)
Duration				
Among current users of CaM				
≤1 year	29 (0.33)	116 (0.28)	1.12 (0.74–1.69)	0.96 (0.62–1.47)
>1 year	14 (0.16)	46 (0.11)	1.36 (0.74–2.48)	1.36 (0.73–2.54)
Among current users of CaD				
≤1 year	199 (2.24)	1040 (2.52)	0.85 (0.73–1.00)	0.84 (0.71–1.00)
>1 year	120 (1.35)	383 (0.93)	1.38 (1.12–1.70)	1.40 (1.11–1.77)
Daily dose of calcium				
Among current users of CaM				
Low dose (<1000 mg/d)	20 (0.23)	95 (0.23)	0.94 (0.58–1.52)	0.88 (0.53–1.45)
High dose (≥1000 mg/d)	14 (0.16)	38 (0.09)	1.67 (0.90–3.09)	1.32 (0.69–2.55)
Unknown	9 (0.10)	29 (0.07)	1.39 (0.65–2.94)	1.23 (0.57–2.68)
Among current users of CaD				
Low dose (<1000 mg/d)	126 (1.42)	647 (1.57)	0.86 (0.71–1.05)	0.83 (0.68–1.03)
High dose (≥1000 mg/d)	145 (1.64)	563 (1.37)	1.14 (0.95–1.38)	1.12 (0.91–1.38)
Unknown	48 (0.54)	213 (0.52)	1.01 (0.74–1.38)	1.05 (0.76–1.46)

* Adjusted only for matching factors (age, sex, and calendar year). ^†^ Adjusted for matching factors (age, sex, and calendar year) and the potential confounding factors shown in Table 1. Except for atrial fibrillation and oral anticoagulant drugs.

## Data Availability

Data not published within this article could be made available by reasonable request from any qualified investigator, provided that the owner of BIFAP (the AEMPS) authorize specifically the data transfer.

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
