# Peer review of "Risk of Ischemic Stroke Associated with Calcium Supplements and Interaction with Oral Bisphosphonates: A Nested Case-Control Study"

_jcm, 2023, doi:10.3390/jcm12165294_

Round 1
Reviewer 1 Report
1) Please clarify the control group selection criteria
2) Table 3. Risk of cardioembolic ischemic stroke associated with the use of calcium supplements- duration users of CaM - correct formation
Author Response
Reviewer #1
- Please clarify the control group selection criteria
Response: Controls came from the underlying cohort and fulfil the same criteria as cases: aged 40 to 99 years, 1-year registry with their primary care physician (PCP), and no previous record of cancer or stroke of any type. We have added this information at the end of the following sentence in Methods (section 2.2):
Five controls per case individually matched with cases by exact age, sex, and index date were randomly selected from the underlying cohort at the time of case occurrence following a risk set sampling. As controls arose from the underlying cohort they fulfilled the same criteria as cases: aged 40 to 99 years, 1-year registry with their primary care physician (PCP), and no previous record of cancer or stroke . Such sampling of controls is incidence-density based (the probability of control selection is proportional to the total person-time at risk), and allows to obtain unbiased estimates of the underlying cohort’s Rate Ratios (RRs) through the ORs computed in the case-control analysis [18].
- Table 3. Risk of cardioembolic ischemic stroke associated with the use of calcium supplements- duration users of CaM - correct formation
Response:. Thanks. We have tried to do our best, but were unable to do this correctly. Hope, it can be refined when edited.
Reviewer 2 Report
The authors analyzed the risk of ischemic stroke when using oral bisphosphonates with calcium supplements. There is a study results that the risk of cardiovascular disease may increase depending on the dose of calcium supplements, and therefore, it is recommended to optimize the intake of calcium and vitamin D for the treatment of osteoporosis. In the past, studies have shown that bisphosphonates may increase the risk of developing atrial fibrillation. The authors also seem to have published a study on the relationship between bisphosphonates and cardioembolic ischemic stroke. Studies on the risk of calcium supplements and cardiovascular disease have yielded conflicting results. Therefore, the research topics of the authors will be of interest to the readers. Overall, the content is adequately written, but a few corrections and supplements are needed.
1. Please describe how the various comorbidities included in the potential confounding factors were defined, or if there is a definition of an existing study, please provide it as a reference. Please also describe whether drugs used together were defined as being used together even if they were prescribed only once or as being used together when they were prescribed for more than a certain period of time.
2. When classifying vascular risk in statistical analysis, why did 1) include diabetes mellitus and atrial fibrillation in establish vascular disease? They are more appropriately classified as 2) vascular risk factors. It is not clinically appropriate to classify PAD, AMI, TIA, diabetes mellitus, and atrial fibrillation as one group. If you reclassify, please correct the figures presented in the table accordingly.
3. In line 169, who is classified as 7) remainder? What is the difference between 1) non-user and 7) remainder?
Author Response
Reviewer #2
The authors analyzed the risk of ischemic stroke when using oral bisphosphonates with calcium supplements. There is a study results that the risk of cardiovascular disease may increase depending on the dose of calcium supplements, and therefore, it is recommended to optimize the intake of calcium and vitamin D for the treatment of osteoporosis. In the past, studies have shown that bisphosphonates may increase the risk of developing atrial fibrillation. The authors also seem to have published a study on the relationship between bisphosphonates and cardioembolic ischemic stroke. Studies on the risk of calcium supplements and cardiovascular disease have yielded conflicting results. Therefore, the research topics of the authors will be of interest to the readers. Overall, the content is adequately written, but a few corrections and supplements are needed.
- Please describe how the various comorbidities included in the potential confounding factors were defined, or if there is a definition of an existing study, please provide it as a reference.
Response: The different comorbidities included as potential confounders are recorded as such in the clinical records by the general practitioners and validated by the BIFAP team through the manual review of a small sample. For some comorbidities treated with specific drugs (diabetes dyslipidemia, angina), we have also used such drugs as disease indicators (as specified). To clarify this further we have added a sentence at the end of the paragraph, as follows:
“To compute the adjusted odds ratios (AOR), the following co-morbidities and other risk factors were considered as possible confounding factors using expert criteria (all of them recorded at the index date or before): Number of visits to primary care physician (PCP) during the year preceding the index date, body mass index (BMI), alcohol abuse, smoking, diabetes (recorded as such and/or by using glucose-lowering drugs as an indicator), hypertension, dyslipidemia (recorded as such and/or by using lipid-lowering drugs as an indicator), hyperuricemia (either asymptomatic or in the context of gout), peripheral artery disease (PAD), acute myocardial infarction (AMI), angina pectoris (recorded as such and/or use of nitrates), transient ischemic attack (TIA), thromboembolic disease, heart failure, chronic kidney failure, rheumatoid arthritis, and chronic obstructive pulmonary disease (COPD). All these comorbidities were recorded as such by the PCPs and further validated by the BIFAP team through the manual review of a small sample. For some comorbidities with specific drugs (diabetes dyslipidemia, angina), we also used such drugs as disease indicators (as specified).
Please also describe whether drugs used together were defined as being used together even if they were prescribed only once or as being used together when they were prescribed for more than a certain period of time.
Response: We confirm that drugs used together are considered as such when either they were used in fixed-dose combinations or when they were concomitantly used as different medicinal products. To clarify this, we have added the following sentence in section 2.4. Exposure definition:
“Among current users, we distinguished those who used them as monotherapy (CaM) or with vitamin D (CaD) (either in fixed-dose combinations or as separate medicinal products)”
- When classifying vascular risk in statistical analysis, why did 1) include diabetes mellitus and atrial fibrillation in establish vascular disease? They are more appropriately classified as 2) vascular risk factors. It is not clinically appropriate to classify PAD, AMI, TIA, diabetes mellitus, and atrial fibrillation as one group. If you reclassify, please correct the figures presented in the table accordingly.
Response: Thanks for this comment. Diabetes mellitus was included among established vascular disease following some reports which suggest that diabetes poses patients to an atherothrombotic risk equivalent to having had a previous acute atherothrombotic event (see Juutilainen A, Lehto S, Ronnemaa T, Pyoraia K, Laakso M. Type 2 diabetes as a “coronary heart disease equivalent”: an 18-year prospective population-based study in Finnish subjects. Diabetes Care. 2005;28:2901–2907). The reason to classify AF among established CV disease is to put emphasis on the importance of AF as an important risk factor for ischemic stroke. However, we understand the reviewer´s view and have followed his/her suggestion. Accordingly, we have amended the Methods sections, the Results text, the table 1, and figures 2 and 3.
- In line 169, who is classified as 7) remainder? What is the difference between 1) non-user and 7) remainder?
Response: The remainder are those patients who do not fulfil the criteria for being non-users (for all drugs), or the criteria for being current user of any of the drugs considered. In other words, the remainder are those patients who are:
- Past users of calcium supplements and past users of oral bisphosphonates
- Past users of calcium supplements and non-users of oral bisphosphonates
- Past users of oral bisphosphonates and non-users of calcium supplements
We have added this clarification in the text as a parenthesis after “7) the remainder”, and also in a footnote in figure 5.